# Profiles of Free and Bound Phenolics and Their Antioxidant Capacity in Rice Bean (*Vigna umbellata*)

**DOI:** 10.3390/foods12142718

**Published:** 2023-07-16

**Authors:** Qinzhang Jiang, Shengwei Wang, Yuzhe Yang, Jinxin Luo, Ruili Yang, Wu Li

**Affiliations:** 1School of Biotechnology and Health Sciences, Wuyi University, Jiangmen 529020, China; jiang3149@126.com (Q.J.); hackfeng0@126.com (S.W.); 2Key Laboratory of Food Nutrition and Functional Food of Hainan Province, College of Food Science and Engineering, Hainan University, Haikou 570228, China; 3College of Food Science, South China Agricultural University, Guangzhou 510642, China; yz3794847@163.com (Y.Y.); luojinxin0124@163.com (J.L.)

**Keywords:** rice bean, free phenolics, bound phenolics, antioxidant

## Abstract

Rice bean (*Vigna umbellata*) is a medicinal and dietary legume rich in polyphenols. In this study, the free and bound phenolics in rice bean were extracted by water, 80% methanol, and acid, base, and composite enzymatic hydrolysis, respectively. The polyphenol profiles of the extracted fractions were analyzed. The outcome demonstrated that base hydrolysis was the most effective way to liberate bound phenolics from rice bean (14.18 mg GAE/g DW), which was 16.68 and 56.72 folds higher than those extracted by acid and enzymatic hydrolysis, respectively. The bound polyphenols released by base hydrolysis contributed to 71.15% of the total phenolic content. A total of 35 individual phenolics was identified, of which isoquercitrin, procyanidin B1, rutin, taxifolin, and catechin were the main monomeric phenolics in the free fraction, while gallic acid, protocatechuic acid, *p*-hydroxybenzoic acid, catechin, and phloroglucinol were the main monomeric phenolics in the bound fraction. In comparison to the free phenolics extracted by water and 80% methanol and the bound phenolics extracted using acid and composite enzymatic hydrolysis, the bound phenolics from base hydrolysis had a superior antioxidant capacity. The antioxidant activity of rice bean is primarily attributed to individual phenolics such as catechin, abundant both in free and bound fractions, and also *p*-hydroxybenzoic acid, gallic acid, and protocatechuic acid in bound fractions. The bound phenolics of rice bean were first reported and showed large differences with the composition of free phenolics. This work suggests that the bound fraction of rice bean must be taken into account in assessing its potential benefits to health.

## 1. Introduction

Polyphenols are a class of secondary metabolites characterized by the presence of multiple phenol structural units that are widely found in nature, and their intake may be associated with the alleviation of chronic and degenerative diseases [1]. Their various pharmacological effects and health benefits such as anti-tumor, anti-atherosclerotic, and anti-inflammatory effects have been revealed in previous studies [2,3,4]. Based on their presence in a matrix, polyphenols can be classified as free or bound polyphenols. Free polyphenols are typically found in fruits, vegetables, and tea [5], and can be easily extracted by solvents of different polarities such as methanol or water [6], due to their major distribution in the vesicles of cells. Bound polyphenols are usually found in grains, nuts, and seeds and form stable covalent bonds with matrix components via esters, ethers, and C-C bonds [7], which accordingly requires extraction methods capable of disrupting molecular structures (e.g., acid hydrolysis, base hydrolysis, and enzymatic hydrolysis) to release them from the matrix [8]. Acid hydrolysis can effectively break the glycosidic bonds and liberate the phenolic compounds from the matrix [9]. Base hydrolysis effectively releases the bound polyphenols from the matrix by breaking ether and ester bonds [10,11]. Enzymatic hydrolysis shows its advantages of being more targeted and gentler in releasing bound phenolics, and different types of enzymes such as pectinase and cellulase can specifically break down the covalent bonds which help release phenolic substances from the matrices of plant cell walls [12]. The efficiencies of different methods to release polyphenols from different matrices varies considerably due to different types of covalent bonds linking phenolics to cell-wall components [13,14].

Pulses are important dietary items in many parts of the world, with the convenience of requiring no shelling while containing more nutrients and phytochemicals than shelled rice [15]. Early research has demonstrated that the seed coat and matrix of pulses contain an abundance of bound phenolics, a major nutritional contributor to the health benefits of pulses [9,16]. Alshikh et al. found that bound phenolics extracted from lentils (*Lens culinaris*) greatly reduced the peroxidation of low-density lipoprotein cholesterol, which had a positive effect on delaying obesity and diabetes development, and the effect of bound phenolics was more pronounced than that of free phenolics [17].

Rice bean (*Vigna umbellata*), also known as red bean and climbing bean, is native to Southeast Asia and widely cultivated in tropical and subtropical regions. Rice bean is a legume of medicinal and food origin and has been used for centuries in Chinese traditional medicine for its diuretic, detoxifying, and pus-draining properties [18]. Recent pharmacological studies have also shown that rice bean can inhibit dipeptidyl peptidase and alpha-glucosidase activity [19], and alleviates acetaminophen-induced liver damage by reducing lipid peroxidation [20]. Polyphenols contribute to the anti-inflammatory and antioxidant activities of rice bean, in which catechin-7-O-β-D-glucopyranoside was reported to lower the expression of tumor necrosis factor-α and interleukin-1β induced by trinitrobenzene sulfonic acid (TNBS), and thus alleviate colitis in rats [21].

Previous studies have also analyzed the free phenolics in rice bean. Yao et al., used 70% ethanol to obtain a high-yield extract of catechin, epicatechin, quercetin, vitexin, isovitexin, *p*-coumaric acid, ferulic acid, and sinapic acid from 13 varieties of Chinese rice bean [22]. Sritongtae et al., applied 95% methanol and water to extract polyphenols from rice bean, and the main individual phenolics were gallic acid, catechin, epicatechin, *p*-coumaric acid, and ferulic acid. It was reported that the total phenolic content and antioxidant activity from the 95% methanolic extract of rice bean were higher than those from the aqueous extraction method [23]. However, these preliminary studies were limited to the free phenolic compounds of rice bean, and the composition, content, and antioxidant power of bound phenolics remain to be explored, which limits the further processing and application of rice bean.

Thus, the focus of this study was to compare several extraction techniques to determine the composition, concentration, and antioxidant properties of free (by water and 80% methanol extraction, respectively) and bound (by acid, base, and composite enzyme hydrolysis, respectively) polyphenols in rice bean. At the same time, the phenolic compounds of rice bean extracted by different methods were characterized and quantified, utilizing ultra-performance LC tandem Q-Exactive Orbitrap mass spectrometry (UPLC-Q-Exactive Orbitrap/MS). Additionally, the antioxidant potency of the free and bound phenolic fractions extracted using various methods was compared. This work tries to provide a comprehensive analytical profile of the composition and content of rice bean polyphenols.

## 2. Materials and Methods

### 2.1. Reagents and Chemicals

Gallic acid, *p*-hydroxybenzoic acid, ferulic acid, trans-cinnamic acid, *p*-coumaric acid, benzoic acid, ellagic acid, protocatechuic acid, sinapic acid, caffeic acid, quercetin, rutin, taxifolin, isoquercitrin, catechin, gallocatechin, epicatechin, vitexin, genistin, daidzin, phloroglucinol, quinic acid, and 3,4-dihdyroxyphenylaceticacid were obtained from Yuanye Bio-Technology Co., Ltd. (Shanghai, China). Procyanidin B1, procyanidin B2, and kaempferol-3-O-rutinoside were obtained from Chengdu Pusi Biotechnology Co., Ltd. (Chengdu, China). 2,20-azinobis (3-ethylbenzothiazoline-6-sulfonic acid (ABTS), 1,1-diphenyl-2-picrylhydrazyl (DPPH), 2,4,6tri(pyridin-2-yl)-1,3,5-triazine (TPTZ), (±)-6-Hydroxy-2,5,7,8-tetramethylchromane-2-carboxylic acid (Trolox), fluorescein, 3′,6′-dihydroxyspiro [isobenzofuran-1 (3 H),9′-(9 H) xanthene]-3-one (FL), 2,2′-azobis (2-amidinopropane) dihydrochloride (AAPH), and EDTA-2Na were purchased from Aladdin Biochemical Technology Co., Ltd. (Shanghai, China). Thermo Fisher Scientific (Waltham, MA, USA) provided the methanol (HPLC quality), formic acid (99.9%, LC-MS grade), and acetonitrile (LC-MS grade). Cellulase (*T richoderma Vride G*, 400 u/mg), hemicellulase (*Aspergillus niger*, 20 u/mg), and pectinase (*Aspergillus niger*, 500 u/mg) were obtained from Yuanye Bio-Technology Co., Ltd. (Shanghai, China).

### 2.2. Plant Material Collection and Preparation

Rice bean (*Vigna umbellata*) was purchased from Zhaoqing Green Agricultural By-Products Distribution Base (origin: Zhaoqing, China). Rice bean was ground to reach a particle size of 0.18–0.25 mm by a grinder (2500A, SUFENG, Yongkang, China) and applied directly in the experiment.

### 2.3. Phenolic Extract

#### 2.3.1. 80% Methanol or Water for Free Phenolics

Free phenolic components from the rice bean were extracted referring to the protocol of Li et al. [24]. As shown in Figure 1, 0.5 g of dry rice bean powder was weighed and combined with 15 mL of either deionized water or an 80% methanol solution containing 1% formic acid. The mixture was ultrasonically sonicated using an ultrasonic cleaner (CR-060S-15L, Chunlin, Shenzhen, China) at 360 W power and 25 °C for 30 min. The supernatant was collected after 15 min of centrifugation at 4 °C and 9000× *g*. In accordance with the aforementioned operation, the above extraction procedures were repeated three times, after which the supernatants were mixed together and a rotary evaporator (N-1300V-WB, EYELA, Tokyo, Japan) was used for concentration at 35 °C in the dark. The dried extract was reconstituted with 5 mL of 80% methanol and kept at −80 °C for analysis. These were the phenolics extracted from water and methanol (W and M). The extraction procedure was repeated in triplicate.

#### 2.3.2. Extraction of Bound Phenolics by the Three Hydrolysis Methods

Acid hydrolysis extraction methods (A) for bound phenolics: The extraction of bound phenolics is shown in Figure 1. After 0.5 g dry weight of rice bean powder was extracted with 80% methanol, the dried residue was hydrolyzed in 15 mL of nitrogen-filled, 3 M concentrated HCl. The mixture was centrifuged at 11,000× *g* for 15 min after being maintained in an 80% water bath for 60 min. The pH of the mixture was then adjusted to 2 with a 10 M NaOH solution. The same volume of ethyl acetate (EA) was applied four times to extract the supernatant. All collected supernatants were concentrated by rotary evaporation in the dark at 35 °C, redissolved in 80% methanol in a fixed volume of 5 mL and then kept at −80 °C before analysis. All extractions were carried out in triplicate.

Base hydrolysis extraction methods (B) for bound phenolics: After 0.5 g dry weight of rice bean powder was extracted with 80% methanol, the dried residue was homogenized with 15 mL of 3 M NaOH solution containing 10 mM EDTA-2Na and 1% ascorbic acid before carrying out the hydrolysis process in a full nitrogen environment. The mixture was stirred in a water bath for 4 h at 30 °C, and the pH was then adjusted to 2 using 6 M HCl. Then, the sample was centrifuged at 11,000× *g* for 15 min at 4 °C. The supernatant was added with the same volume of ethyl acetate (EA) for extraction; the procedure was repeated four times, and the collected extracts were combined. All of the supernatants were concentrated by rotary evaporation in the dark at 35 °C, redissolved in 80% methanol in a fixed volume of 5 mL, and then kept at −80 °C before analysis.

Composite enzyme hydrolysis (E) for bound phenolics: After 0.5 g dry weight of rice bean powder was extracted with 80% methanol, the dried residue was combined with 0.03 g of composite enzymes (cellulase:hemicellulose:pectinase = 1:1:1), and 15 mL of water (pH-adjusted to 5.0 using citric acid) was added to the combination. The mixture was then incubated for two hours at 50 °C in a shaking water bath. After enzymatic hydrolysis, the mixtures were maintained at 80 °C for 10 min to deactivate the enzymes and then underwent a 30 min ultrasonic extraction process at 50 °C with 360 W of ultrasonic power. After being swiftly brought to room temperature in an ice bath, the liquid was centrifuged for 15 min at 11,000× *g*. The supernatant was added with the same volume of ethyl acetate (EA) for extraction which was repeated four times, and the collected extracts were combined. All of the supernatants were concentrated by rotary evaporation in the dark at 35 °C, redissolved in 80% methanol in a fixed amount of 5 mL, and then kept at −80 °C before analysis. All extractions were performed in triplicate.

### 2.4. Total Phenolic and Total Flavonoid Content Determination

The concentration of all phenolic compounds was determined using the Folin–Ciocalteu reagent [25]. In a nutshell, 150 µL of the extracts were combined with 0.5 mL of distilled water for 6 min at room temperature with 150 µL of Folin–Ciocalteu reagent. Then, 1.25 mL of 7% sodium carbonate solution was added. The solution’s absorbance at 760 nm was measured following a 90 min incubation period at 30 °C. The results of each extract were checked three times. The standard reference (R^2^ = 0.999) was gallic acid (10~100 g/mL). The TPC value of the samples was expressed as the gallic acid equivalents per gram of dry weight.

Little modification was made to the previously reported aluminum trichloride method [25] in order to measure the level of total flavonoids. Then, 50 µL of NaNO_2_ (5%, *w*/*v*) solution was combined with 100 µL of methanol and 100 µL of extracts and incubated for 6 min. Then, 50 µL of a 10% (*w*/*v*) AlCl_3_ solution was added for stirring thoroughly for another 6 min. In order to stop the process, 400 µL of NaOH (1 M) and 300 µL of methanol were added. Following a 15 min incubation period at room temperature, the mixture’s absorbance at 510 nm was gauged. The results of each extract were checked three times. The standard reference used was rutin (50~600 g/mL; R^2^ = 0.997). The TFC value of the samples was expressed as the rutin equivalents per gram of dry weight.

### 2.5. Antioxidant Activity

The DPPH free radical scavenging experiment was carried out mainly in accordance with the procedure described by He et al. [26]. An amount of 50 µL of the sample was combined with 400 µL of DPPH methanolic solution (100 mM), and the mixture was held at room temperature for 30 min without exposure to light. Utilizing a microplate reader (Synergy Neo2, BioTek, Winooski, VT, USA), absorbances were measured at 517 nm for 200 µL of each solution sample held in a 96-well plate. The blank control was methanol. Every test was run in triplicate. The standard used was Trolox (10~150 µg/mL; R^2^ = 0.995). The results of the samples were quantified in terms of the Trolox equivalents per gram of dry weight.

With a few minor modifications, the described approach in [27] was applied to perform the ABTS + experiment. In a nutshell, 10 mL of ABTS + solution (7.0 mM) and 176 µL of potassium persulfate solution (140 mM) were combined. After being incubated at 4 °C for 16 h in the dark, the stock solution was diluted with distilled water until the absorbance at 734 nm was 0.7 ± 0.02. An amount of 100 µL of the sample was mixed with 2 mL of the diluted ABTS+ stock solution and then left at room temperature for 6 min without being exposed to light. Each solution (200 µL) was poured to a 96-well plate, and a microplate reader was used to measure the absorbance at 734 nm. All tests were performed in triplicate. The standard reference used was Trolox (10~100 µg/mL; R^2^ = 0.995). The results of the samples were quantified in terms of the Trolox equivalents per gram of dry weight.

A minorly modified classic methodology described by Benzie et al. [28] was applied for the ferric reducing antioxidant power (FRAP) assay. The acetate buffer (300 mM, pH 3.6), 10 mM TPTZ (dissolved in 40 mM HCl), and FeCl_3_ (20 mM) were combined in a ratio of 10:1:1 (*v*/*v*/*v*) to prepare the FRAP reagent. The reagents were freshly manufactured and warmed in a water bath at 37 °C. A 30 µL sample was combined with 900 µL of FRAP solution, and the mixture was held at room temperature in the dark for 30 min. The solution’s absorbance was determined at 593 nm for samples on a microplate reader. Every test was run in triplicate. The standard used was FeSO_4_·7H_2_O (0~800 µmol/mL; R^2^ = 0.999). The results of the samples were quantified in terms of the ferrous sulfate equivalents per gram of dry weight.

The assay of oxygen radical absorbance capacity (ORAC) was performed as previously described in [29] with slight modifications. A PBS buffer (0.1 mol/L, pH 7.4) was used to prepare the sodium fluorescein solution and AAPH solution. An amount of 25 µL of the sample or Trolox (6.25~500 µmol/mL) was poured to a black microtiter plate. A working buffer containing 100 µL of 150 nmol/L sodium fluorescein was added, and the mixture was equilibrated at 37 °C with periodic shaking for 20 min. The system was then filled with 75 µL of 119.4 mM AAPH, and the plate was placed within a microplate fluorometer. (Fluoroskan Ascent, Thermo LabSystems, Franklin, MA, USA). The reaction was automatically mixed and started according to a set time. The excitation wavelength was set to 485 nm and the absorption wavelength to 535 nm, and the fluorescence intensity was measured once every 3.5 min. The area of fluorescence burst was calculated from the fluorescence intensity decay curve. The final ORAC values of the samples were calculated using Trolox as the reference compound (R^2^ = 0.999), where the area was proportional to the Trolox concentration. The results of the samples were quantified in terms of the Trolox equivalents per gram of dry weight.

### 2.6. UPLC-Q-Exactive Orbitrap MS/MS Analysis

A UPLC-Q-Exactive Orbitrap MS/MS (Thermofisher Scientific, Shanghai, China) coupled to an electrospray ionization (ESI) source was applied to identify phenolic components. Phenolics were determined as described previously in [30] with liquid chromatogram conditions: Waters ACQUITY UPLC BEH C18 column (2.1 × 100 mm, 1.7 µm). The mobile phases A and B were Milli-Q grade water containing 0.1% formic acid and acetonitrile, respectively. The gradient elution profile was set as follows: 0–3 min 5–15% B, 3–11 min 15–30% B, 11–15 min 30–50% B, 15–21 min 50–90% B, 21–22 min 90% to 5% B. The injection volume was 2 µL, and the flow rate was 0.15 mL/min. The MS conditions were set as follows: capillary voltage was held at 3200 V, capillary temperature was 320 °C, resolution was 7000, and the resolution of the sheath gas flow of nitrogen was 35 arb, and the aux gas flow was 10 arb. In the range of 70–1050 *m*/*z*, positive and negative ion scanning modes of mass spectra (MS)/spectra were observed.

A comparison of the spectra and retention periods of the extracts with externally injected standards was applied for the identification of phenolics in the extracts: gallic acid, quercetin, ellagic acid, quinic acid, protocatechuic acid, rutin, isoquercitrin, catechin, gallocatechin, epicatechin, vitexin, genistin, sinapic acid, daidzin, caffeic acid, ferulic acid, trans-cinnamic acid, *p*-coumaric acid, benzoic acid, phloroglucinol, *p*-hydroxybenzoic acid, 3,4-dihydroxyphenylacetic acid, taxifolin, kaempferol-3-O-rutinoside, procyanidin B1, and procyanidin B2. By comparing the precise mass of the parent ion (M-H) with the regular MS fragmentation pattern, compounds with no available standards were identified according to the references. Each phenolic component’s content in the extracts was determined using standards. External calibration curves for each standard were created for quantification.

### 2.7. Statistical Analysis

Each experiment was carried out in triplicate, and the findings were shown as the mean ± standard deviation. SPSS was employed for statistical analyses. To compare the mean values of the experimental results from different extraction methods, an ANOVA test (Tukey’s and Bonferroni) was conducted. The significance of statistical differences was set at *p*-values equal to or less than 0.05.

## 3. Results and Discussion

### 3.1. Different Procedures for the Extraction of Free and Bound Polyphenols

As shown in Figure 2, extractions by 80% methanol and distilled water extracts generated total phenolic contents of 5.75 mg GAE/g DW and 3.67 mg GAE/g DW, respectively. Also, a flavonoid content of 4.43 mg RE/g DW was obtained through 80% methanol extraction, whereas for the water extraction, the extracted flavonoid value was 3.30 mg RE/g DW. The results showed that the free polyphenol and free flavonoid contents in rice bean released by the 80% methanol extraction method were higher than those of the water extraction method.

Polyphenols are soluble in both water and a variety of organic solvents. Organic solvents such as ethanol and methanol are most usually used for the extraction of polyphenols. However, due to differences in the polarity of solvents, the solubility of different types of polyphenols varies [31]. Our results show that polyphenols from rice bean are more soluble in methanol compared to water. Therefore, it can be concluded from the results of this study that methanol can better facilitate the extraction of free phenolic compounds from rice bean compared to water.

As shown in Figure 2, the hydrolysis procedures significantly differ in how effectively bound phenolics were released. In contrast to the acid hydrolysis (0.85 mg GAE/g DW) and composite enzymatic hydrolysis (0.25 mg GAE/g DW) extracts, the base hydrolysis extract contained much higher levels of bound phenolics, reaching 14.18 mg GAE/g DW, 16.68 folds higher than that from the acid hydrolysis and 56.72 folds higher than that from the composite enzyme hydrolysis. Similarly, the base hydrolysis extract contained 2.85 mg RE/g DW of bound flavonoids, 11.88 and 14.25 folds higher than those from the acid hydrolysis (0.24 mg RE/g DW) and composite enzyme hydrolysis (0.20 mg RE/g DW), respectively. At the same time, the bound phenolics and flavonoids extracted from the base hydrolysis accounted for 71.15% and 39.15% of the total phenolic and total flavonoid content from the 80%-methanol-based extraction and 79.44% and 46.34% of the total phenolic and total flavonoid content from the water-based extraction. These results suggest that bound phenolics contribute to the majority of polyphenols extracted from rice bean and that rice bean is a potential source of polyphenols.

Our results indicated that, of the three extraction methods, the release of rice bean bound phenolics was significantly higher from the base hydrolysis extraction than from the acid and composite enzyme hydrolysis methods. These findings are consistent with those of earlier research in which base hydrolysis was more efficient in releasing bound phenolics from mung bean hulls and soybeans compared to acid hydrolysis, respectively [14,32]. Most of the bound phenolics in beans are mainly bound to the cell wall through ether and ester bonds [33], while another small proportion is bound through glycosidic bonds, which may explain why base hydrolysis is more effective in releasing bound phenolics in rice bean. However, base hydrolysis is not always an efficient method for the extraction of plant bound phenolics. Joanna et al. [13] found that acid hydrolysis extractions from cocoa beans contained more catechin, epicatechin, ellagic acid, procyanidin B2, and n-phenylpropenoyl-l-amino acids compared to base hydrolysis extractions. These studies show contrastingly different efficiencies in the extraction of bound phenolics when comparing the tested hydrolysis methods’ applicability to different matrices.

### 3.2. Profiles of Individual Phenolic Compounds in Various Extracts

By using the UPLC-Q-Exactive Orbitrap MS/MS, the phenolic components (free and bound phenolics) extracted from rice bean were identified and quantified. In Table 1, the phenolics’ characteristic compositions are displayed. According to the accurate mass parent ion peak, retention time, and fragmentation ion pattern, a total of 35 phenolic compounds was identified and categorized, including 6 hydroxybenzoic acid derivatives, 5 hydroxycinnamic acid derivatives, 20 flavonoids, and 4 kinds of other phenolics.

Compounds 15, 26, and 27 were identified by externally injected standards, compounds 11, 13, 18, 28, 29, 30, 33, 34, and 35 by references, and the other 23 compounds were identified by both standards and references [34,35,36,37,38,39,40]. The corresponding identification information is shown in the Appendix A.

### 3.3. Individual Phenolic Compounds Quantitative in Different Extracts

After the extraction by different extraction methods, free and bound phenolics were released from rice bean. Table 2 shows the contents of individual phenolic compounds in different extracted fractions. Overall, among all the extracted fractions, isoquercitrin (2.63–163.45 mg/kg DW), procyanidin B1 (3.35–106.63), catechin (0.14–72.04), rutin (0.68–58.07), taxifolin (0.17–52.85), *p*-hydroxybenzoic acid (0.53–38.85), and protocatechuic acid (0.76–23.58) were the most abundant phenolic compounds extracted from rice bean. In terms of free phenolics, the main phenolic compounds from the deionized water extract were isoquercitrin (87.96), procyanidin B1 (68.12), catechin (53.35), and taxifolin (52.85). The main phenolic compounds from the 80% methanol extract were isoquercitrin (163.45), procyanidin B1 (106.63), rutin (58.07), and catechin (13.18). In terms of bound phenolics, catechin (72.04), *p*-hydroxybenzoic acid (38.85), phloroglucinol (28.26), protocatechuic acid (23.58), procyanidin B1 (17.58), gallic acid (13.42), and 3,4-dihydroxyphenylacetic acid (6.86) were obtained through the base hydrolysis extraction of rice bean. The main phenolic components from the acidic hydrolysis extracts were *p*-hydroxybenzoic acid (20.93), protocatechuic acid (19.40), and gallic acid (9.75). Benzoic acid (21.86), catechin (4.86), and *p*-coumaric acid (4.18) constituted the major phenolic composition from the composite enzymatic extracts.

The quantitative results show that different solvents for extraction have a significant effect on the release of phenolic compounds from rice bean. Procyanidin B1, isoquercitrin, and rutin were 1.57, 1.86, and 4.69 folds higher in the 80% methanol extract than in the water extraction, while taxifolin and catechin were 3.04 and 4.04 folds higher in the water extract than in the 80% methanol extraction, respectively. In addition, genistin and daidzin were only present in the extract from the 80% methanol fraction. Phenolic acids and flavonoids are the main phenolic compounds found in beans [41,42]. The significant differences in the content of flavonoids observed in the present work may be due to the polarity of the extraction solvent and the different solubility of the phenolics among different solvents. Most flavonoids are very sensitive to enzymatic and non-enzymatic oxidation, while organic solvents can inhibit both enzymatic and non-enzymatic oxidation [43]. The addition of methanol releases more of the poorly water-soluble rutin from rice bean, and the methanol–water combination extracted higher levels of procyanidin B1 and isoquercitrin than from the water extract alone. However, in terms of relative polarity, the binary solvent system was less effective in the extraction of certain phenolics as compared to the single solvent system; the content of catechin was higher in the water extraction than in the 80% methanol extraction based on the present work. The discovery is identical to the finding of Erol et al. [44], in which water was more efficient in extracting catechin from green tea compared to methanol and ethanol. Taxifolin and quercetin were also more efficiently extracted by water in this study.

Compared to the acid and composite enzymatic extractions, the base hydrolysis method exhibits the greatest potential to release bound phenolic compounds in rice bean, such as catechin (514.57 folds higher than in the acid extract and 14.92 folds higher than in the enzymatic extract) and gallocatechin (46.42 folds higher than in the acid extract and 8.70 folds higher than in the enzymatic extract), as well as procyanidin B1, procyanidin B2, phloroglucinol, and 3,4-dihydroxyphenylacetic acid were not found in the acid and enzyme extract fractions. Compared to the free phenolic extract fractions, flavonoids such as procyanidin B1, isoquercitrin, and rutin were significantly reduced in the bound phenolic fractions. However, the phenolic acids in rice bean were released in large quantities by the three hydrolysis methods applied in this study. Bound phenolic compounds are normally covalently bound to the insoluble macromolecules of the plant and therefore require a strong or even destructive extraction method for their release from the matrix [9]. According to previous studies, phenolic acids are the most prevalent bound phenolic compounds in cereals and legumes [45]. In this work, the results showed that bound phenolics in rice bean contained a large number of phenolic acids which were significantly higher than the free phenolics. This is consistent with the results of Duenas et al. [46] who found 12 phenolic acids in black beans, most of which were bound phenolic acids. Among the three groups of bound phenolic fractions, gallic acid, protocatechuic acid, *p*-hydroxybenzoic acid, and *p*-coumaric acid were found to be more abundant in the base hydrolysis fraction, suggesting that most phenolic acids in rice bean are bound to macromolecules through ether and ester bonds. This observation is in line with the findings of Zheng et al. [32] who found that base hydrolysis was more efficient than acid hydrolysis in the extraction of bound phenolics from mung bean skin. Previous research has demonstrated that bound phenolic acids can be ether-bonded to lignin monomers and are often esterified to cell wall polysaccharides [47]. In addition, the low phenolic acid content of the acid hydrolysis fraction may be due to high temperatures during the hydrolysis process, which is consistent with the previous study [48]. Composite enzyme hydrolysis is an emerging method for the extraction of bound phenols in recent years, but its mildness with lower-level structural destruction compared to acid and base hydrolysis makes this hydrolysis method very ineffective in bound phenolic extraction from rice bean, with only benzoic acid extracted more efficiently than that from acid or base hydrolysis fractions, which could be possibly ascribed to the fact that benzoic acid is usually combined with the cellulose plant matrix [49]. It is worth noting that the base hydrolysis fraction is also rich in bound flavonoids such as catechin and procyanidin B1. The phenolic groups in catechin can be donors for hydrogen bonds, and multiple hydrogen bonds enable polyphenols to bind strongly to macromolecules such as proteins and lipids, which may also account for the richness of procyanidin B1 and catechin in the base hydrolysis fraction [50].

### 3.4. Antioxidant Activity of Phenolics in Rice Bean

In this study, ABTS+, DPPH radical scavenging assays, and FRAP and ORAC assays were performed to assess the antioxidant activity of free phenolics and bound phenolics released from rice bean after different extraction protocols. As shown in Table 3, the free phenolic extracts exhibited a favorable antioxidant capacity, and compared to the water extract (14.48 µmol TE/g DW for ABTS+, 11.96 µmol TE/g DW for DPPH, 18.99 µmol Fe(II)SE/g DW for FRAP, and 45.96 µmol TE/g DW for ORAC), the methanolic extract had stronger antioxidant activity with values of 26.71 µmol TE/g DW for ABTS+, 17.96 µmol TE/g DW for DPPH, 22.87 µmol Fe(II)Se/g DW for FRAP, and 74.36 µmol TE/g DW for ORAC.

Among the bound phenolic extracts, the base hydrolysis fraction showed the strongest antioxidant activity, with 90.49 µmol TE/g DW for ABTS+, 75.74 µmol TE/g DW for DPPH, 73.04 µmol Fe(II)SE/g DW for FRAP, and 161.09 µmol TE/g DW for ORAC, significantly higher than that from the acid hydrolysis (4.94 µmol TE/g DW for ABTS+, 6.60 µmol TE/g DW for DPPH, 3.77 µmol Fe(II)SE/g DW for FRAP, and 16.29 µmol TE/g DW for ORAC) and composite enzyme hydrolysis (0.89 µmol TE/g DW for ABTS+, 0.36 µmol TE/g DW for DPPH, 0.97 µmol TE/g DW for FRAP, and 1.72 µmol TE/g DW for ORAC).

A previous study reported the high antioxidant activity of free phenolic compounds in 13 different varieties of rice bean [22]. However, the antioxidant capacity of the bound phenolics in rice bean has not been elucidated. The antioxidant activity of plant extracts is also closely related to the phenolic compounds they contain [51]. The results of the current study show that the base hydrolysis fraction has the highest polyphenol contents. Consistently, extract fractions with a high phenolic content have a strong antioxidant capacity.

The Pearson correlation coefficients between TPC, TFC, DPPH, ABTS+, FRAP, and ORAC and the main monocot phenolics of the rice bean extracts are shown in Table 4. The total phenolic content was positively and significantly correlated with the antioxidant activity (r = 0.983, 0.993, 0.996, 0.997 for DPPH, ABTS+, FRAP, ORAC, respectively). In previous studies, the antioxidant activity of 16 legumes, including rice bean, showed a significantly positive correlation with total phenolic content [52]. However, there was no significant positive correlation detected between TFC and the antioxidant activity of rice bean in this work. Flavonoids, with the exception of catechin, were mainly present in a free form in rice bean, and taking into account the substantial amount of phenolic acids in the bound form, our finding show that bound phenolic acids extracted from rice bean make an important contribution to antioxidant capacity.

The contribution of certain phenolics in rice bean to the overall antioxidant potential, however, has not been confirmed. In the current study, the antioxidant capacity of the phenolic extract of rice bean was further examined, as well as the contributions of the key individual phenolic components. The relative antioxidant capacities of specific phenolics of rice bean appear to be phloroglucinol > catechin > *p*-hydroxybenzoic acid > gallic acid > protocatechuic acid. In addition, it can be noted in the results that, apart from catechin, flavonoids such as taxifolin, isoquercitrin, quercetin, procyanidin B1, and rutin, which are more abundant in free phenolics, do not seem to contribute in terms of the antioxidant capacity of rice bean, which may be due to the high antioxidant capacity of the base extract bound phenolic fraction and its lower flavonoid content compared to the free phenolic fraction. The present work show that flavonoids contribute predominantly to the antioxidant activity of free phenolics, while it is mainly phenolic acids that contribute to the antioxidant capacity of the bound phenolics in rice bean.

## 4. Conclusions

In this study, the composition, contents, and antioxidant activity of free and bound phenolics in rice bean were investigated for the first time. The results show that bound phenols are the main components of rice bean polyphenols. Catechin, taxifolin, isoquercitrin, quercetin, procyanidin B1, and rutin are the main free phenolic compounds in rice bean, and catechin, *p*-hydroxybenzoic acid, phloroglucinol, gallic acid, protocatechuic acid, and benzoic acid are the main bound phenolics in rice bean. The total phenolic content of rice bean showed a significant positive correlation with antioxidant capacity. Individual phenolics such as catechin, rich in free and bound fractions, and phloroglucinol, *p*-hydroxybenzoic acid, gallic acid, and protocatechuic acid in the bound fraction contribute to the main antioxidant activity of rice bean. These results suggest that the bound fraction must be taken into account when assessing rice bean’s potential benefits to health. In contrast to free polyphenols, bound polyphenols escape the effects of gastrointestinal digestion and are released, transformed, and exert their active effects during colonic fermentation [53]; therefore, rice bean rich in bound polyphenols will be a potential source of natural antioxidants.

## Figures and Tables

**Figure 1 foods-12-02718-f001:**
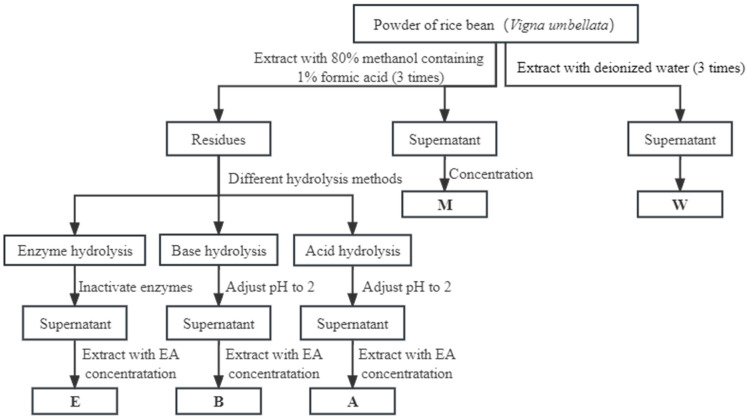
Procedure for the extraction of free and bound phenolics from rice bean: EA, ethyl acetate; W, free phenolics extracted by water; M, free phenolics extracted by 80% methanol; A, bound phenolics extracted by acid hydrolysis; B, bound phenolics extracted by base hydrolysis; E, bound phenolics extracted by composite enzyme hydrolysis.

**Figure 2 foods-12-02718-f002:**
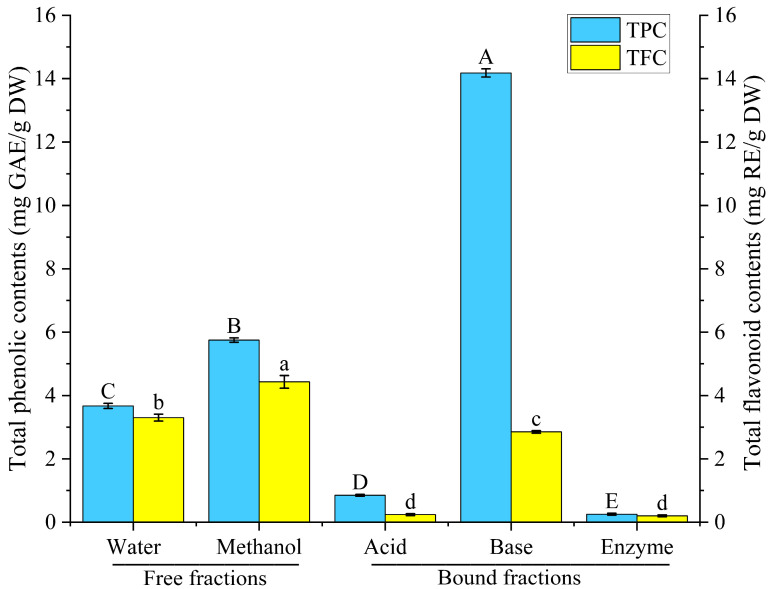
The total phenolic and flavonoid contents of rice bean released using different methods. TPC: total phenolic contents; TFC: total flavonoid contents. Different letters indicate significant differences in total phenolic (capital letters) or total flavonoid (lower case letters) contents for different extraction methods, *p* < 0.05.

**Table 1 foods-12-02718-t001:** Identification of individual phenolic compounds in rice bean in different phenolic fractions.

Peak.	Formula	Compounds	RT (Min)	*m*/*z* Parents	*m*/*z* Fragments	Fractions
	**Hydroxybenzoic acids**					
3	C_7_H_6_O_5_	gallic acid ^ab^	2.97	169.013	125.023	W, M, A, B, E
5	C_7_H_6_O_4_	protocatechuic acid ^ab^	4.98	153.018	109.028	W, M, A, B, E
9	C_7_H_6_O_3_	*p*-hydroxybenzoic acid ^ab^	6.47	137.023	93.033	W, M, A, B, E
13	C_9_H_10_O_5_	syringic acid ^b^	7.21	197.045	182.022, 166.998	W, M, B, E
17	C_14_H_6_O_8_	ellagic acid ^ab^	8.89	300.999	257.009	W, M, A, B, E
27	C_7_H_6_O_2_	benzoic acid ^a^	11.90	121.028	94.028, 122.032	W, M, A, B, E
	**Hydroxycinnamic acids**					
12	C_9_H_8_O_4_	caffeic acid ^ab^	7.10	179.034	135.044	W, M, B, E
19	C_9_H_8_O_3_	*p*-coumaric acid ^ab^	8.97	163.039	119.049	W, M, A, B, E
22	C_11_H_12_O_5_	sinapic acid ^ab^	9.52	223.061	208.037, 193.013	W, M, B, E
23	C_10_H_10_O_4_	ferulic acid ^ab^	9.67	193.050	178.026, 149.060	W, M, A, B, E
32	C_9_H_8_O_2_	trans-cinnamic acid ^ab^	15.62	147.044	102.947	W, M, A, B, E
		**Flavonoids**				
4	C_15_H_14_O_7_	gallocatechin ^ab^	4.42	305.067	261.077, 179.034	W, M, A, B, E
7	C_30_H_26_O_12_	procyanidin B1 ^ab^	5.55	577.135	451.103, 407.077	W, M, B, E
8	C_15_H_14_O_6_	catechin ^ab^	6.28	289.072	245.081, 179.034	W, M, A, B, E
10	C_30_H_20_O_12_	procyanidin B2 ^ab^	6.61	577.135	451.105, 407.077	W, M, B, E
14	C_15_H_14_O_6_	epicatechin ^ab^	7.24	289.072	245.081, 179.034	W, M, B
15	C_21_H_20_O_9_	daidzin ^a^	7.81	415.104	253.051	M, A, B
16	C_27_H_30_O_16_	rutin ^ab^	8.84	609.146	300.028	W, M, B
18	C_21_H_20_O_10_	isovitexin ^b^	8.95	431.098	341.067, 311.056	W, M
20	C_21_H_20_O_10_	vitexin ^ab^	9.06	431.098	311.056, 341.066	W, M, A, B
21	C_21_H_20_O_12_	isoquercitrin ^ab^	9.35	463.088	300.027, 301.035	W, M, B
24	C_21_H_20_O_10_	genistin ^ab^	9.81	431.099	269.045, 268.038	M
25	C_27_H_30_O_15_	kaempferol-3-0-rutinoside ^ab^	9.93	593.152	285.040	W, M, B
26	C_15_H_12_O_7_	taxifolin ^a^	10.07	303.506	259.071, 241.062	W, M, A, B, E
28	C_15_H_10_O_7_	morin ^b^	13.19	301.036	273.040, 255.030	W, M, A
29	C_15_H_10_O_4_	daidzein ^b^	13.53	253.051	135.009, 107.133	W, M
30	C_16_H_12_O_5_	glycitein ^b^	13.96	283.061	268.037	W, M, A
31	C_15_H_10_O_7_	quercetin ^ab^	14.53	301.035	178.998, 151.003	W, M, A
33	C_15_H_10_O_5_	genistein ^b^	16.13	269.046	133.028	W, M
34	C_15_H_12_O_5_	naringenin ^b^	16.19	271.061	151.003, 119.049	W, A, B
35	C_15_H_10_O_6_	kaempferol ^b^	16.39	285.041	257.046	W, M, B
		**Others**				
1	C_7_H_12_O_6_	quinic acid ^ab^	1.68	191.055	127.039	E
2	C_6_H_6_O_3_	phloroglucinol ^ab^	2.82	127.039	109.029	B
6	C_8_H_8_O_4_	3,4-dihydroxyphenylaceticacid ^ab^	5.49	167.034	123.044	B
11	C_6_H_6_O_2_	catechol ^b^	6.71	109.028	91.018	M, A

^a^ compared with standards, ^b^ compared with references. W: water extracts; M: 80% methanol extracts; A: acid hydrolysis extracts; B: base hydrolysis extracts; E: enzyme hydrolysis extracts.

**Table 2 foods-12-02718-t002:** Contents of individual phenolic compounds in rice bean in different phenolic fractions (mg/kg DW).

No.	Compound	Free Fractions		Bound Fractions		
		W	M	A	B	E
**Hydroxybenzoic acids and derivatives**						
1	gallic acid	1.05 ± 0.03 d	4.14 ± 0.18 c	9.75 ± 0.42 b	13.42 ± 0.55 a	0.46 ± 0.02 e
2	protocatechuic acid	5.60 ± 0.24 c	6.84 ± 0.16 c	19.40 ± 1.03 b	23.58 ± 2.27 a	0.76 ± 0.08 d
3	*p*-hydroxybenzoic acid	2.56 ± 0.32 cd	3.94 ± 0.14 c	20.93 ± 1.70 b	38.85 ± 2.06 a	0.53 ± 0.07 d
4	ellagic acid	4.89 ± 1.56 b	8.06 ± 2.33 a	5.52 ± 1.92 ab	3.16 ± 0.14 bc	1.32 ± 0.05 c
5	benzoic acid	2.42 ± 0.77 b	1.70 ± 0.26 bc	1.16 ± 0.12 bc	0.80 ± 0.05 c	21.86 ± 1.08 a
**Hydroxycinnamic acids and derivatives**						
6	caffeic acid	0.02 ± 0.00 d	0.08 ± 0.01 c	NF	0.23 ± 0.01 a	0.16 ± 0.03 b
7	*p*-coumaric acid	2.40 ± 0.33 c	1.58 ± 0.34 d	1.49 ± 0.07 d	4.82 ± 0.37 a	4.18 ± 0.18 b
8	sinapic acid	0.98 ± 0.13 a	0.59 ± 0.05 b	NF	0.21 ± 0.01 c	0.08 ± 0.02 c
9	ferulic acid	0.99 ± 0.13 a	0.60 ± 0.06 b	0.03 ± 0.01 c	0.14 ± 0.01 c	0.51 ± 0.05 b
10	trans-cinnamic acid	0.76 ± 0.05 b	1.06 ± 0.17 a	0.71 ± 0.04 b	0.74 ± 0.05 b	0.72 ± 0.06 b
	**Flavonoids**					
11	catechin	53.35 ± 5.51 b	13.18 ± 0.68 c	0.14 ± 0.17 d	72.04 ± 6.14 a	4.83 ± 0.21 d
12	taxifolin	52.85 ± 6.13 a	17.41 ± 0.49 b	0.30 ± 0.04 c	0.76 ± 0.02 c	0.17 ± 0.03 c
13	epicatechin	0.20 ± 0.02 b	0.20 ± 0.03 b	NF	5.61 ± 0.44 a	NF
14	isoquercitrin	87.97 ± 7.99 b	163.45 ± 3.68 a	NF	2.63 ± 0.34 c	NF
15	quercetin	24.58 ± 1.65 a	4.24 ± 0.22 b	0.08 ± 0.07 c	NF	NF
16	gallocatechin	2.95 ± 0.27 b	1.51 ± 0.48 c	0.12 ± 0.00 d	5.57 ± 0.45 a	0.64 ± 0.01 d
17	procyanidin B1	68.12 ± 5.79 b	106.63 ± 6.15 a	NF	17.58 ± 0.53 c	3.35 ± 0.15 d
18	procyanidin B2	1.98 ± 0.19 b	2.05 ± 0.11 b	NF	6.16 ± 0.48 a	0.37 ± 0.02 c
19	rutin	12.37 ± 1.35 b	58.07 ± 3.15 a	NF	0.68 ± 0.14 c	NF
20	kaempferol-3-0-rutinoside	1.31 ± 0.12 b	4.02 ± 0.13 a	NF	0.27 ± 0.00 c	NF
21	daidzin	NF	0.28 ± 0.04 c	1.37 ± 0.07 b	1.97 ± 0.37 a	NF
22	genistin	NF	0.25 ± 0.07 a	NF	NF	NF
23	vitexin	0.03 ± 0.00 a	0.03 ± 0.01 a	0.02 ± 0.01 b	0.01 ± 0.00 b	NF
	**Others**					
24	quinic acid	NF	NF	NF	NF	0.27 ± 0.04 a
25	phloroglucinol	NF	NF	NF	28.26 ± 0.19 a	NF
26	3,4-dihydroxyphenylaceticacid	NF	NF	NF	6.86 ± 0.60 a	NF
	**Total**	**327.38 ± 32.58 b**	**399.91 ± 18.94 a**	**61.02 ± 5.67 d**	**234.34 ± 9.22 c**	**40.21 ± 2.10 d**

The mean ± standard deviation is the unit expressed for values. NF: not found; W: water extracts; M: 80% methanol extracts; A: acid hydrolysis extracts; B: base hydrolysis extracts; E: enzyme hydrolysis extracts; Different lowercase letters indicate significant differences in polyphenol content between the extraction methods (*p* < 0.05).

**Table 3 foods-12-02718-t003:** The ABTS+, DPPH radical scavenging activity, ferric reducing antioxidant activity, and oxygen radical absorbance capacity of phenolics released from rice bean following different extractions.

Stage	Free Fractions		Bound Fractions		
	W	M	A	B	E
ABTS+ radical scavenging activity (µmol TE/g DW)	14.48 ± 0.05 c	26.71 ± 0.74 b	4.94 ± 0.21 d	90.49 ± 3.01 a	0.89 ± 0.05 e
DPPH radical scavenging activity (µmol TE/g DW)	11.96 ± 0.60 c	17.96 ± 0.58 b	6.60 ± 0.22 d	75.74 ± 2.38 a	0.36 ± 0.01 e
Ferric reducing/antioxidant power (µmol Fe(II)SE/g DW)	18.99 ± 0.35 c	22.87 ± 0.62 b	3.77 ± 0.09 d	73.04 ± 0.67 a	0.97 ± 0.04 e
Oxygen radical absorbance capacity (µmol TE/g DW)	45.96 ± 6.23 c	74.36 ± 7.43 b	16.29 ± 1.39 d	161.09 ± 1.97 a	1.72 ± 0.34 e

The mean ± standard deviation is the unit expressed for values. (a–e): there are statistically significant differences between the antioxidant capacities of the various extracts, as indicated by different lowercase letters in the same row (*p* < 0.05).

**Table 4 foods-12-02718-t004:** Analysis of the Pearson correlation coefficient between the antioxidant activity and phenolics in rice bean that were extracted using various processes.

	TPC	TFC	DPPH	ABTS	FRAP	ORAC
TPC	1.000	0.541 *	0.983 **	0.993 **	0.996 **	0.997 **
TFC	0.541 *	1.000	0.382	0.440	0.496	0.593 *
catechin	0.812 **	0.51	0.794 **	0.789 **	0.845 **	0.797 **
taxifolin	−0.106	0.556 *	−0.218	−0.200	−0.095	−0.075
*p*-hydroxybenzoic acid	0.744 **	−0.044	0.839 **	0.801 **	0.765 **	0.727 **
isoquercitrin	0.008	0.828 **	−0.172	−0.103	−0.056	−0.076
quercetin	−0.127	0.447	−0.216	−0.208	−0.103	−0.106
phloroglucinol	0.929*	0.190	0.977 **	0.962 **	0.946 *	0.897 *
procyanidin B1	0.121	0.895 **	−0.061	0.007	0.062	0.185
gallic acid	0.669 **	−0.024	0.754 **	0.721 **	0.675 **	0.668 **
protocatechuic acid	0.616 *	−0.041	0.707**	0.665 **	0.632*	0.617 *
rutin	0.049	0.752 **	−0.118	−0.045	−0.031	0.116
benzoic acid	−0.487	−0.565 *	−0.445	−0.442	−0.472	−0.543 *

TPC: total phenolic content; TFC: total flavonoid content; * correlation was significant at the 0.05 level (two-tailed); ** correlation was significant at the 0.01 level (two-tailed).

## Data Availability

Not applicable.

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
