# Peer review of "Profiles of Free and Bound Phenolics and Their Antioxidant Capacity in Rice Bean (Vigna umbellata)"

_foods, 2023, doi:10.3390/foods12142718_

Round 1
Reviewer 1 Report
The corrections suggested for the manuscript are:
- Page 1 in abstract, line 6 - After "enzymatic hydrolysis" add ", respectively".
- Page 1 and 2 - Check reference "[2]". Perhaps, replace reference "[2]" with a more adequate one or correct these sentences.
- Page 2 - Replace reference "[19]" with a more adequate or correct this sentence and replace stated flavonoids with phenolic compounds "catechin, epicatechin, p-coumaric acid, ferulic acid, vitexin, isovitexin, sinapic acid, quercetin" which are analyzed in Rice Bean in the study "[19]".
- Page 4 - Check reference "[22]". Is it reference "[21]"?
- Page 5 and 6 - If possible; add reference for "UPLC-Q-Exactive Obitrap MS/MS Analysis".
- Typos correction:
Page 9 - Change "Siringic acid" to "siringic acid", "C7H6O3" to "C7H6O3" and after "cinnamic acid" add ".".
Page 11 - Change "zheng" to "Zheng".
Reviewer 2 Report
Grammar, punctuation, and capitalization of words contains extensive errors. Please have a native English-speaking person edit your manuscript.
Section 2.3.2- EDTA-2NA is not listed in the materials section and is not defined.
In section 3.2 the ion m/z numbers should not have 4 or 5 significant figures after the decimal point. It is not necessary for the interpretation of the data, and mass spectrometry is not accurate to that many decimal places.
Also in section 3.2 the chemical formulas should have the numbers in subscript.
The quality of the English language is not very good and detracts from the science in the paper.
Reviewer 3 Report
The work is focused on the study of the phenolic profile, both free and bound phenolic compounds, from rice bean, investigated through different extraction methods. The novelty of the work is based on the study of the bound phenolic fraction, which has been concluded to be an important nutritional source of antioxidant compounds. Hence, I recommend this work, but a general revision of English spelling is recommended as well as more keywords could be included to intensify the importance of the work. Also, the following minor corrections should be made:
Introduction:
- Please give a reference for “and can be easily extracted by sol-vents of different polarity such as methanol or water.”
- Please give a reference for “…requiring extraction methods capable of disrupting the structure (e.g. acid hydrolysis, base hydrolysis and enzymatic hydrolysis) to release them from the matrix.”
- Check the grammar in: “Acid hydrolysis effectively breaking the glycosidic bonds and allowing the phenolic com-pounds to be leached from the matrix [6].”
- Explain deeply and give a possible explanation for: “Enzymatic hydrolysis is a more targeted and gentle method of releasing bound phenolics [9].”
- More review about research background on extraction from rice bean is missed. For example: methods used, main extracted compounds…
Materials and Methods:
- Section 2.2: The results are expressed by dry sample so, what is the moisture of the rice bean samples?
- Section 2.3.1: “…the extract were reconstituted with 5 mL of 80% methanol and stored at -80ºC for determination.” Is the water extract also reconstituted in 80% metOH?
- Section 2.3.2: “The dried residue (0.5 g) generated after methanol extracted was hydrolyzed…”, 0.5g is the initial amount of sample used for metOH extraction. Check.
- Check the spelling for “volumn”
- It is said that for acid hydrolysis pH was adjusted to pH 2 with NaOH 10M, is correct?
- Section 2.6: What about the gradient from 11-15 min?
Results and Discussion:
- Check the spelling in Figure 2.
- Specify the statistical analysis in the legend of Figure 2.
- Figure 2B is a repetition of the data shown in Fig2A, I recommend to just discuss it in the text but not to duplicate the information in the figures.
- Check the name of the section 3.3.
- Check this sentence: “The total phenolic content was significantly and positively correlated with the antioxidant activity (r=0.979, 0.991, 0.995, 0.998 for DPPH, ABTS+, FRAP, ORAC, respectively).” These results seem to not agree with those shown in table 4.
A general revision of english spelling should be made through all the manuscript: grammar and word's spelling.
Reviewer 4 Report
The manuscript entitled “Profiles of free and bound phenolics and their antioxidant capacity in rice bean (Vigna umbellata)|” is of interest.
Several statements are repeated throughout the introduction section, this must be avoided to make the paper more comprehensive.
It is important to clearly state the implications for research, practice, and society. Healthy choices must include foods with nutrients and bioactive compounds for a balance diet, I kindly recommend the next papers to be consulted: http://dx.doi.org/10.5772/intechopen.91218
Avoid repetition and try to focus on the main ideas regarding the results. At the same time, the Results and Discussions section could be improved by studying other papers in the field.
The conclusion section - I recommend the addition of relevant ideas from the current study. In my opinion, I think readers of this paper may want to know about the future trends of rice bean-based products. It would be good if the authors could base it on the existing literature to give a more solid forecast about the trends.
It is important to analyze the grammar structure and punctuation.
